# The Impact of Coagulation Biomarkers on Survival Outcomes in Adult Glioblastoma

**DOI:** 10.3390/medicina61040756

**Published:** 2025-04-19

**Authors:** Rahmi Atıl Aksoy, Timur Koca, Yasemin Şengün, Ece Atak, Aylin Fidan Korcum

**Affiliations:** 1Department of Radiation Oncology, Akdeniz University Faculty of Medicine, Antalya 07070, Turkey; timurkoca@akdeniz.edu.tr (T.K.); yaseminsengunn@gmail.com (Y.Ş.); eceatak.md@gmail.com (E.A.); aylinkorcum@akdeniz.edu.tr (A.F.K.); 2Department of Radiation Oncology, Izmir City Hospital, Izmir 35530, Turkey; 3Department of Radiation Oncology, Van Education and Research Hospital, Van 65300, Turkey

**Keywords:** glioblastoma, coagulation, prothrombin time, D-dimer, albumin, survival

## Abstract

*Background and Objectives*: Glioblastoma presents a significant challenge in oncology due to its aggressive nature and poor prognosis, despite advancements in treatment. This study aims to comprehensively evaluate the prognostic significance of coagulation biomarkers, including the novel albumin/D-dimer ratio, in adult glioblastoma patients. *Material and Methods*: This retrospective study included 74 adult glioblastoma patients who underwent Stupp protocol treatment. Blood samples were collected before radiotherapy to measure biomarkers, including prothrombin time (PT), activated partial thromboplastin time (aPTT), albumin, D-dimer, and the albumin/D-dimer ratio. The prognostic significance of these biomarkers for progression-free survival (PFS) and overall survival (OS) was assessed using both univariate and multivariate Cox regression analyses. *Results*: The median follow-up time was 12.2 months (range, 1–77.4 months). Univariate analysis revealed that ECOG performance status (*p* = 0.001), D-dimer (*p* = 0.03), and albumin (*p* = 0.001) were significant prognostic factors for PFS. Multivariate analysis identified albumin (*p* = 0.02) as an independent prognostic biomarker for PFS. For OS, univariate analysis showed that age (*p* = 0.004), ECOG performance status (*p* = 0.001), tumor volume (*p* = 0.007), extent of resection (*p* = 0.01), radiotherapy dose (*p* = 0.001), D-dimer (*p* = 0.02), albumin (*p* = 0.001), albumin/D-dimer ratio (*p* = 0.02), and PT (*p* = 0.002) were significant prognostic factors. Multivariate analysis revealed age (*p* = 0.04), extent of resection (*p* = 0.02), and PT (*p* = 0.04) as independent prognostic factors for OS. *Conclusions*: Our findings highlight the prognostic significance of coagulation biomarkers, particularly PT, D-dimer, albumin, and the albumin/D-dimer ratio, in glioblastoma. These biomarkers may serve as valuable tools for prognostic assessment and personalized treatment strategies, warranting further exploration in larger prospective studies.

## 1. Introduction

Glioblastoma, the most aggressive and prevalent primary malignant brain tumor, poses a significant challenge in neuro-oncology due to its poor prognosis and limited therapeutic advancements [1]. Despite advancements in multimodal treatment approaches, including maximal safe resection, radiotherapy, and temozolomide-based chemotherapy known as the Stupp protocol, median overall survival remains poor and rarely exceeds two years [2,3]. Established prognostic factors such as patient age, performance status, extent of resection, and molecular markers (e.g., IDH mutation status and MGMT promoter methylation) are widely recognized [4]. However, glioblastoma’s marked heterogeneity and complex tumor biology highlight the necessity for additional biomarkers to refine prognostic stratification and support personalized treatment strategies.

Emerging evidence underscores the intricate relationship between systemic inflammation and coagulation in cancer progression. These interconnected processes drive crucial aspects of tumor biology, including angiogenesis, immune evasion, and metastatic potential [5]. In glioblastoma, inflammatory biomarkers such as the neutrophil-to-lymphocyte ratio (NLR) and platelet-to-lymphocyte ratio (PLR) have demonstrated prognostic significance [6,7]. Moreover, composite indices like the Pan-Immune Inflammation Value (PIV) and Global Immune-Nutrition-Inflammation Index (GINI) have recently emerged as promising tools, further linking systemic inflammation to survival outcomes [8,9].

Coagulation, a hallmark of cancer, plays a critical role in tumor progression and metastasis. Tumor cells interact with the coagulation system through multiple mechanisms, including tissue factor expression, the release of procoagulant microparticles, and inflammatory cytokine production, which collectively contribute to a hypercoagulable state [10,11]. This prothrombotic environment not only increases the risk of venous thromboembolism (VTE) but also facilitates tumor progression. Notably, glioblastoma patients exhibit an exceptionally high VTE incidence, with a two-year cumulative incidence of 7.5% and a 30% increase in mortality risk [12,13].

A study reported that glioblastoma patients often exhibit a hypercoagulable profile, characterized by reduced prothrombin time (PT) and activated partial thromboplastin time (aPTT), along with elevated levels of D-dimer and von Willebrand factor (VWF), all of which are associated with adverse survival outcomes [14]. D-dimer, a fibrin degradation product, has gained attention as a biomarker of hypercoagulability and tumor aggressiveness, with elevated levels linked to both an increased risk of VTE and poorer survival [15,16]. Similarly, serum albumin, a well-established marker of nutritional status and systemic inflammation, has been independently associated with survival across multiple malignancies, including glioblastoma [17,18]. Despite these findings, the prognostic value of coagulation parameters remains underexplored compared to inflammatory biomarkers. Moreover, composite indices, such as the albumin/D-dimer ratio, have demonstrated prognostic significance in other malignancies; however, their relevance in glioblastoma remains inadequately studied and warrants further investigation [19,20].

This study aims to comprehensively investigate the prognostic significance of coagulation biomarkers and novel indices, such as the albumin/D-dimer ratio, in adult glioblastoma patients. By analyzing these biomarkers, we seek to clarify their role in prognostic assessment and clinical decision-making, ultimately contributing to a more personalized and evidence-based approach to glioblastoma management.

## 2. Materials and Methods

### 2.1. Patient Selection

This study was approved by the Clinical Research Ethics Committee of Akdeniz University Faculty of Medicine (Decision number: 728, Date: 13 October 2021) and conducted in accordance with the principles of the Declaration of Helsinki. Due to its retrospective design, the requirement for informed consent was waived. Additionally, the study followed the REMARK guidelines to ensure methodological rigor [21].

A total of 74 patients with histologically confirmed glioblastoma who underwent postoperative radiotherapy plus concurrent and adjuvant temozolomide at our tertiary care hospital between January 2014 and July 2021 were included in this study. Eligibility criteria required patients to be ≥18 years old and have an Eastern Cooperative Oncology Group performance status (ECOG-PS) of 2 or lower. Patients were excluded if they had another primary tumor or metastasis, a disease affecting the coagulation system, regular use of anticoagulant medication, or unavailable blood test results. Additionally, patients with acute or chronic infections, autoimmune or hematological disorders, chronic liver or renal conditions, or those on medications that could influence complete blood count parameters were excluded.

### 2.2. Treatment Protocol and Follow-Up Procedures

External beam radiotherapy was administered to all patients within 4 weeks after surgery. Sixty-six patients received a total of 60 Gy (2.0 Gy/day in 30 fractions), while eight patients received a total of 40 Gy (2.66 Gy/day in 15 fractions). During radiotherapy, temozolomide was administered at a daily dose of 75 mg/m^2^. Adjuvant temozolomide was initiated 4 weeks after the completion of radiotherapy at a dose of 150–200 mg/m^2^ for 5 days every 28 days. Although 6 cycles of adjuvant temozolomide were planned, the number of cycles was adjusted based on tolerability and radiologic response. Supplementary medications, including antiepileptics and steroids, were prescribed only when clinically indicated. All patients underwent routine clinical and radiographic evaluations during follow-up visits. Patients who missed these visits were contacted by telephone at the data cut-off point to determine their final status.

### 2.3. Data Collection

Clinical and pathological characteristics, along with laboratory parameters, were retrospectively collected from the hospital archives. Patient characteristics, including age, gender, ECOG-PS, tumor volume, tumor location, and extent of resection, were documented. Tumor volume was determined using the contrast-enhancing component on T1-weighted sequences of preoperative contrast-enhanced brain magnetic resonance imaging (MRI). Tumor location was categorized by laterality (left, right, or bilateral) and anatomical region (frontal, parietal, temporal, or occipital). The extent of resection was classified as subtotal resection or near/gross total resection (N/GTR), based on postoperative contrast-enhanced brain MRI and surgical notes. Pathological characteristics and molecular markers, including the Ki-67 index and IDH mutation status, were assessed. Laboratory parameters, including D-dimer, PT, aPTT, albumin levels, and platelet counts were recorded prior to the initiation of radiotherapy. The albumin/D-dimer ratio was calculated by dividing the albumin level by the D-dimer level.

### 2.4. Statistical Analyses

All statistical analyses were performed using SPSS, version 24.0 (IBM Corp., Armonk, NY, USA). *p* values < 0.05 were considered statistically significant. Patient characteristics were summarized using frequencies for categorical variables and by using medians and ranges for continuous variables. Receiver operating characteristic (ROC) curve analysis was performed to evaluate the sensitivity and specificity of the variables for predicting overall survival (OS), and cut-off values were determined using Youden’s Index.

Progression-free survival (PFS) was calculated from the end of radiotherapy until progression or the last follow-up, and OS was calculated from the end of radiotherapy until death or the last follow-up. The impact of variables on PFS and OS was assessed using univariate Cox regression analyses, with hazard ratios (HR) and 95% confidence intervals (CIs) reported. Variables showing a *p*-value of less than 0.05 in the univariate analyses were subsequently included in the multivariate Cox regression models. Survival curves were constructed using the Kaplan–Meier method, and the differences between the survival curves were examined using the log-rank test.

Furthermore, Spearman’s correlation coefficient was employed to evaluate the relationships between tumor volume and laboratory parameters.

## 3. Results

Patient characteristics are summarized in Table 1. The median age was 59 years (range, 23–82 years) and most of the patients were male (62.2%). ECOG-PS was 0–1 in 56 patients, and 2 in 18 patients. The most affected lobes were the frontal (35.1%) and parietal (35.1%) lobes, followed by the temporal lobe (24.4%) and the occipital lobe (5.4%). The left hemisphere was affected in 32 cases (43.2%), the right hemisphere in 36 (48.6%). Both hemispheres were affected in six (8.2%) cases. The median tumor volume was 38.8 cm^3^ (range, 2.9–120.1 cm^3^). N/GTR was performed in 33 cases (44.6%), subtotal resection in 41 cases (55.4%). During the treatment period, VTEs were documented in 11 patients (14.9%). Deep vein thrombosis was detected in five patients, pulmonary thromboembolism in two patients, and both in four patients. The median follow-up period of patients was 12.2 months (range, 1–77.4 months).

ROC curve analyses are illustrated in Figure 1. The cut-off values for predicting OS were determined based on the areas under the curve (AUC) in the ROC analyses, as follows: D-dimer—0.55 μg/mL; albumin—4.3 g/dL; albumin/D-dimer ratio—6.5; PT—12.53 s; aPTT—23 s; platelet count—313 × 10^3^; and tumor volume—36.3 cm^3^. The D-dimer at 0.55 μg/mL had a sensitivity of 85% and a specificity of 63% (*p* = 0.008); albumin at 4.3 g/dL had a sensitivity of 74% and a specificity of 98% (*p* = 0.001); albumin/D-dimer ratio at 6.5 had a sensitivity of 81% and a specificity of 73% (*p* = 0.007); PT at 12.53 s had a sensitivity of 66% and a specificity of 85% (*p* = 0.004); aPTT at 23 s had a sensitivity of 51% and a specificity of 79% (*p* = 0.29); platelet count at 313 × 10^3^ had a sensitivity of 67% and a specificity of 53% (*p* = 0.45); and tumor volume at 36.3 cm^3^ had a sensitivity of 70% and a specificity of 90% (*p* = 0.004).

The 1- and 2-year PFS rates of patients were 36% and 24%, respectively, with a median PFS of 7.0 months (95% CI: 3.2–10.8 months). Patients with D-dimer < 0.55 μg/mL had a significantly longer median PFS than those with D-dimer ≥ 0.55 μg/mL (9.4 vs. 2.4 months; *p* = 0.03) (Figure 2a). No significant difference in median PFS was observed between patients with PT < 12.53 s and those with PT ≥ 12.53 s (7.2 vs. 6.0 months; *p* = 0.21) (Figure 2b). Univariate analysis demonstrated lower PFS for patients with ECOG-PS 2 (*p* = 0.001), higher D-dimer level (*p* = 0.03), and lower albumin level (*p* = 0.001). In multivariate analysis, albumin level (*p* = 0.02) remained a significant prognostic variable for PFS (Table 2).

The 1- and 2-year OS rates of patients were 50% and 33%, respectively, with a median OS of 11.9 months (95% CI: 6.8–16.9 months). Patients with D-dimer < 0.55 μg/mL had a longer median OS than those with D-dimer ≥ 0.55 μg/mL (16.5 vs. 9.1 months; *p* = 0.02) (Figure 3a). Additionally, median OS in patients with PT ≥ 12.53 s was longer than that in those with PT < 12.53 s (19.0 vs. 7.7 months; *p* = 0.002) (Figure 3b). Univariate analysis demonstrated lower OS for patients with older age (*p* = 0.004), ECOG-PS 2 (*p* = 0.001), larger tumor volume (*p* = 0.007), subtotal resection (*p* = 0.01), lower RT dose (*p* = 0.001), higher D-dimer level (*p* = 0.02), lower albumin level (*p* = 0.001), lower albumin/D-dimer ratio (*p* = 0.02), and lower PT (*p* = 0.002). In multivariate analysis, age (*p* = 0.04), extent of resection (*p* = 0.02), and PT (*p* = 0.04) remained significant prognostic variables for OS (Table 3).

Furthermore, a positive correlation was observed between tumor volume and D-dimer levels (r = 0.44, *p* = 0.03). In contrast, tumor volume showed a negative association with albumin levels (r = −0.45, *p* = 0.007) and the albumin/D-dimer ratio (r = −0.46, *p* = 0.03). No statistically significant correlation was found between tumor volume and either PT or aPTT.

## 4. Discussion

Thrombosis and bleeding are among the leading causes of morbidity and mortality in cancer patients, second only to the malignancy itself, and are strongly associated with poorer survival outcomes and reduced quality of life. Tumors disrupt the hemostatic balance by releasing pro-coagulant factors, which promote abnormal clot formation and activate the coagulation system. This process creates a self-perpetuating cycle that not only exacerbates thrombotic risk but also drives tumor progression [22]. Recent studies have highlighted the prognostic significance of coagulation biomarkers across various malignancies, emphasizing their role in predicting survival and guiding therapeutic strategies [23,24,25]. Specifically, glioblastoma cells have been shown to express and secrete coagulation factor X (FX), which actively promotes thrombin generation, further contributing to the prothrombotic state in these patients [26]. Moreover, in gliomas, coagulation factors such as thrombin and tissue factor play a crucial role in tumor invasion and metastasis by activating protease-activated receptors (PARs) in the tumor microenvironment. This activation facilitates key processes, including angiogenesis, migration, and interaction with host vascular cells, all of which contribute to tumor progression [27]. Building on these established associations, our study aims to further investigate the prognostic role of coagulation biomarkers in glioblastoma. Specifically, we focus on pre-radiotherapy coagulation biomarkers in adult glioblastoma patients treated with the Stupp protocol. Our findings establish a significant correlation between these biomarkers and survival outcomes, highlighting their potential clinical relevance in refining risk stratification strategies. Importantly, this study represents a comprehensive evaluation of coagulation biomarkers in glioblastoma and introduces the albumin/D-dimer ratio as a novel composite biomarker for this patient population.

D-dimer is a degradation product of fibrin, and its negative result is frequently utilized as a diagnostic criterion to exclude the presence of VTE [28,29]. Elevated D-dimer levels have been associated with poor prognosis in various solid tumors [30]. In a pilot study including 23 glioblastoma patients, D-dimer levels > 1 μg/mL were shown to correlate with poorer PFS and OS compared to D-dimer levels ≤ 1 μg/mL [15]. In a study by Koudriavtseva et al., D-dimer levels were evaluated in glioma, multiple sclerosis, and control groups. The results revealed significantly higher D-dimer levels in glioma patients compared to both controls and multiple sclerosis patients. Moreover, D-dimer levels ≥ 0.18 μg/mL in glioma patients were associated with worse OS [16]. Beyond serum D-dimer levels, elevated D-dimer concentrations in cerebrospinal fluid have been suggested as a potential marker for spinal dissemination in glioblastoma patients [31]. Consistent with these findings, our study also observed that D-dimer levels ≥ 0.55 μg/mL were associated with poorer PFS and OS compared to D-dimer levels < 0.55 μg/mL. Furthermore, the positive correlation between tumor volume and D-dimer levels suggests that increased tumor volume in glioblastoma may be linked to heightened activation of both coagulation and fibrinolytic systems.

Serum albumin is a crucial marker for assessing visceral protein function. Both malnutrition and inflammation can negatively affect albumin synthesis, resulting in reduced serum albumin levels. In adults, the normal reference range for serum albumin is between 3.5 and 5.0 g/dL, with values below 3.5 g/dL classified as hypoalbuminemia. Albumin has also been recognized as a valuable prognostic biomarker in various cancers [17,32]. Moreover, pre-treatment serum albumin levels have been identified as prognostic indicators in glioblastoma [33,34]. A study involving 214 glioblastoma patients demonstrated that albumin levels < 3 g/dL were associated with worse OS compared to levels ≥3 g/dL [18]. Similarly, our study found that albumin levels < 4.3 g/dL were linked to poorer PFS and OS in glioblastoma patients. Furthermore, a negative correlation was observed between tumor volume and albumin levels, suggesting that larger tumor volumes may reflect heightened systemic inflammation in glioblastoma patients. The mechanisms underlying the association between hypoalbuminemia and poor outcomes warrant further investigation, particularly in the context of systemic inflammation and tumor biology.

Beyond the individual prognostic significance of albumin and D-dimer, our study also explored the prognostic value of the albumin/D-dimer ratio in glioblastoma. Emerging evidence suggests that albumin/D-dimer ratio serves as a valuable prognostic biomarker across various malignancies. In gastric cancer, a lower albumin/D-dimer ratio (<41.6) has been associated with reduced disease control and poorer survival outcomes [19]. Similarly, in advanced lung adenocarcinoma, patients with low albumin/D-dimer ratios (≤16.6) exhibited significantly lower disease control and overall response rates, as well as worse PFS [20]. Despite these findings, no prior studies have investigated the prognostic relevance of albumin/D-dimer ratio in glioblastoma, highlighting a critical gap in the literature. Our study is the first to address this gap by demonstrating that an albumin/D-dimer ratio < 6.5 is significantly associated with poorer OS in glioblastoma patients. Given the well-established interplay between coagulation, inflammation, and tumor progression, albumin/D-dimer ratio may serve as a novel and clinically relevant prognostic biomarker in glioblastoma.

PT and aPTT are simple, cost-effective, and widely used laboratory tests that assess coagulation, anticoagulation, and fibrinolysis. Emerging evidence suggests that these parameters hold significant prognostic value in various malignancies. Elevated PT and aPTT levels have been consistently associated with poor prognosis in gastrointestinal cancers, including colorectal cancer and hepatocellular carcinoma [35,36]. Additionally, elevated PT levels have been identified as a prognostic biomarker for postoperative recurrence in patients with stage I–III colorectal cancer following radical surgery [37]. Conversely, in glioblastoma, the relationship between coagulation parameters and prognosis follows an inverse pattern. Reduced PT and aPTT levels, indicative of a hypercoagulable state, have been linked to adverse survival outcomes [14]. Furthermore, thrombocytosis, commonly associated with decreased PT levels, has also been identified as a marker of poor prognosis in glioblastoma patients [38]. Consistent with these findings, our study demonstrated that low PT levels are associated with worse OS in glioblastoma. However, the lack of a significant correlation between tumor volume and PT or aPTT suggests that tumor size may not directly influence these coagulation parameters. This could indicate that other molecular or systemic factors, such as the presence of inflammatory cytokines, vascular abnormalities, or treatment effects, may be contributing to the regulation of coagulation pathways in glioblastoma.

While this study provides valuable insights into the prognostic significance of coagulation biomarkers in adult glioblastoma patients undergoing the Stupp protocol, several limitations must be acknowledged. First, the retrospective, single-center design and the relatively small sample size of 74 patients may limit the generalizability of the findings. Although patients with known conditions affecting blood parameters were excluded, the potential influence of undetected confounders remains, as peripheral blood cell counts can be affected by various factors, including infections and nutritional status. Furthermore, although these coagulation biomarkers are significant in glioblastoma prognosis, it is important to consider that their patterns may indicate internal brain bleeding, possibly reflecting recent trauma or prior surgery. Similarly, the lack of molecular testing, including IDH mutation status and MGMT promoter methylation, limits the interpretability of our findings. Moreover, the absence of fibrinogen data—a key coagulation marker—prevents a more comprehensive assessment of its prognostic value. Future studies should integrate fibrinogen measurements and explore the interplay between molecular markers and coagulation parameters to enhance our understanding of their prognostic significance in glioblastoma.

## 5. Conclusions

This study underscores the important prognostic role of biomarkers, specifically D-dimer, PT, and albumin levels, in adult glioblastoma. Elevated D-dimer levels were linked to poorer survival outcomes, while higher PT and albumin levels were associated with better outcomes. The albumin/D-dimer ratio, a novel composite biomarker, may provide additional prognostic value. These results highlight the potential of incorporating biomarkers into clinical decision-making, offering a path toward more personalized treatment strategies for glioblastoma patients. Future research should focus on understanding the mechanistic relationship between coagulation pathways and glioblastoma progression, clarifying how hypercoagulability contributes to tumor behavior. The impact of anticoagulant therapies on glioblastoma treatment outcomes also warrants further investigation to assess their clinical relevance. Larger prospective studies are needed to validate these findings and evaluate the clinical applicability of these biomarkers in glioblastoma management.

## Figures and Tables

**Figure 1 medicina-61-00756-f001:**
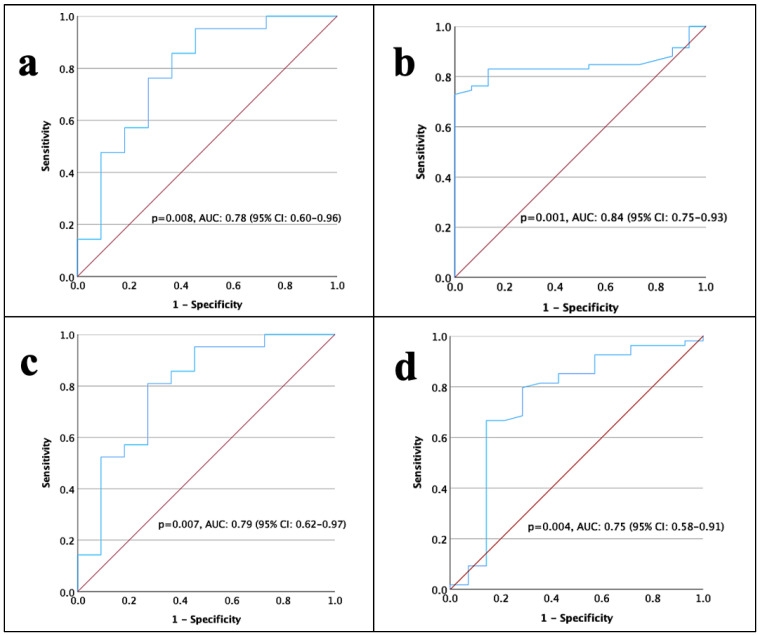
Receiver Operating Characteristic (ROC) curve analysis evaluating laboratory parameters in adult glioblastoma. (**a**) D-dimer. (**b**) Albumin. (**c**) Albumin/D-dimer ratio. (**d**) Prothrombin Time.

**Figure 2 medicina-61-00756-f002:**
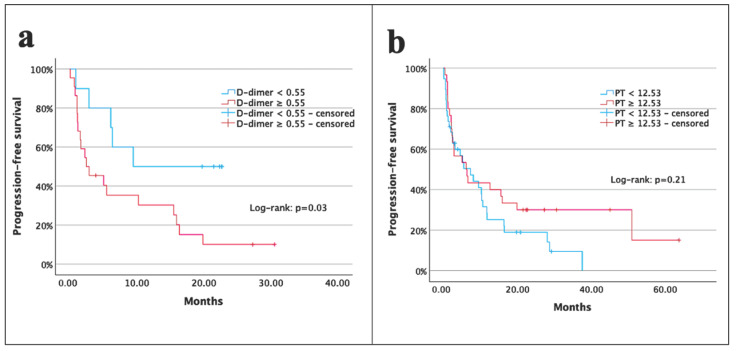
Kaplan–Meier survival curves illustrating progression-free survival based on D-dimer levels (**a**) and prothrombin time (**b**).

**Figure 3 medicina-61-00756-f003:**
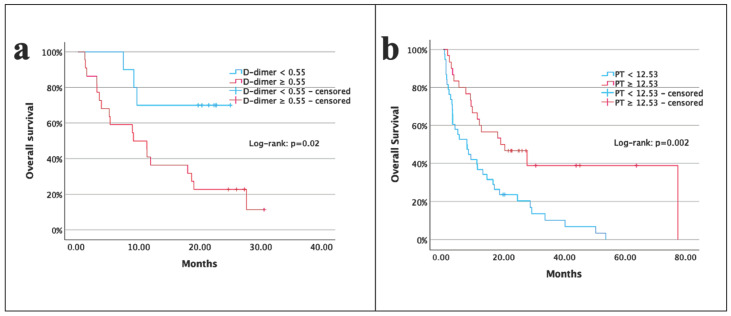
Kaplan–Meier survival curves illustrating overall survival based on D-dimer levels (**a**) and prothrombin time (**b**).

**Table 1 medicina-61-00756-t001:** Patient characteristics (n = 74).

Characteristic	n	%
Age (years)		
<65	49	66.2
≥65	25	33.8
Gender		
Female	28	37.8
Male	46	62.2
ECOG-PS		
0	24	32.4
1	32	43.2
2	18	24.4
Location		
Frontal	26	35.1
Parietal	26	35.1
Temporal	18	24.4
Occipital	4	5.4
Laterality		
Right	36	48.6
Left	32	43.2
Bilateral	6	8.2
Extent of resection		
Subtotal resection	41	55.4
N/GTR	33	44.6
IDH mutation status		
Wild	45	60.8
Mutant	8	10.8
Unknown	21	28.4
Ki-67 index		
<30%	28	37.8
≥30%	24	32.4
Unknown	22	29.8
Total radiotherapy dose		
40 Gy	8	10.8
60 Gy	66	89.2

Abbreviations: ECOG-PS—Eastern Cooperative Oncology Group performance status; N/GTR—near/gross total resection; IDH—isocitrate dehydrogenase.

**Table 2 medicina-61-00756-t002:** Univariate and multivariate Cox proportional hazard analysis of potential prognostic factors for progression-free survival.

Variables	Cut-Off	Univariate Analysis		Multivariate Analysis	
		HR (95% CI)	*p*	HR (95% CI)	*p*
Age (years)	<65 vs. ≥65	1.7 (0.9–3.0)	0.06		
Gender	Male vs. Female	0.8 (0.5–1.4)	0.54		
ECOG-PS	0–1 vs. 2	2.8 (1.5–5.1)	0.001	0.4 (0.1–1.6)	0.24
Tumor volume	<36.3 vs. ≥36.3	1.9 (0.8–4.2)	0.11		
EoR	Subtotal vs. N/GTR	0.7 (0.4–1.1)	0.15		
Ki-67 index	<30% vs. ≥30%	1.01 (0.7–1.3)	0.96		
RT dose (Gy)	60 vs. 40	1.7 (0.7–4.0)	0.20		
D-dimer (μg/mL)	<0.55 vs. ≥0.55	2.8 (1.0–7.6)	0.03	1.9 (0.6–5.9)	0.24
Albumin (g/dL)	<4.3 vs. ≥4.3	0.3 (0.2–0.6)	0.001	0.2 (0.1–0.8)	0.02
Albumin/D-dimer	<6.5 vs. ≥6.5	0.5 (0.2–1.2)	0.12		
PT (s)	<12.53 vs. ≥12.53	0.7 (0.3–1.1)	0.21		
aPTT (s)	<23 vs. ≥23	0.6 (0.4–1.1)	0.12		
Platelet count (10^3^)	<313 vs. ≥313	0.7 (0.3–1.1)	0.20		

Abbreviations: CI—confidence interval; HR—hazard ratio; ECOG-PS—Eastern Cooperative Oncology Group performance status; EoR—extent of resection; N/GTR—near/gross total resection; RT—radiotherapy; PT—prothrombin time; aPTT—activated partial thromboplastin time.

**Table 3 medicina-61-00756-t003:** Univariate and multivariate Cox proportional hazard analysis of potential prognostic factors for overall survival.

Variables	Cut-Off	Univariate Analysis		Multivariate Analysis	
		HR (95% CI)	*p*	HR (95% CI)	*p*
Age (years)	<65 vs. ≥65	2.1 (1.2–3.7)	0.004	6.5 (1.2–35.1)	0.04
Gender	Male vs. Female	0.8 (0.5–1.5)	0.65		
ECOG-PS	0–1 vs. 2	2.5 (1.4–4.5)	0.001	0.3 (0.1–14.6)	0.58
Tumor volume	<36.3 vs. ≥36.3	3.4 (1.4–8.3)	0.007	4.1 (0.5–33.3)	0.17
EoR	Subtotal vs. N/GTR	0.4 (0.2–0.8)	0.01	0.2 (0.1–0.7)	0.02
Ki-67 index	<30% vs. ≥30%	1.2 (0.8–1.6)	0.23		
RT dose (Gy)	60 vs. 40	3.5 (1.6–7.7)	0.001	0.2 (0.1–2.9)	0.14
D-dimer (μg/mL)	<0.55 vs. ≥0.55	3.9 (1.1–13.4)	0.02	2.3 (0.1–44.8)	0.57
Albumin (g/dL)	<4.3 vs. ≥4.3	0.2 (0.1–0.4)	0.001	0.5 (0.1–9.5)	0.69
Albumin/D-dimer	<6.5 vs. ≥6.5	0.2 (0.1–0.8)	0.02	3.9 (0.3–44.4)	0.27
PT (s)	<12.53 vs. ≥12.53	0.4 (0.2–0.7)	0.002	0.2 (0.1–0.8)	0.04
aPTT (s)	<23 vs. ≥23	0.6 (0.3–1.0)	0.09		
Platelet count (10^3^)	<313 vs. ≥313	0.6 (0.3–1.1)	0.13		

Abbreviations: CI—confidence interval; HR—hazard ratio; ECOG-PS—Eastern Cooperative Oncology Group performance status; EoR—extent of resection; N/GTR—near/gross total resection; RT—radiotherapy; PT—prothrombin time; aPTT—activated partial thromboplastin time.

## Data Availability

The data that support the findings of this study are available from the corresponding author upon reasonable request.

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
