# Peer review of "The Impact of Coagulation Biomarkers on Survival Outcomes in Adult Glioblastoma"

_medicina, 2025, doi:10.3390/medicina61040756_

Round 1

Reviewer 1 Report

Comments and Suggestions for Authors

This is an interesting article about the impact of coagulation biomarkers on survival outcomes in
adult glioblastoma.

The introduction section sets the background of the current study as it briefly describes glioblastoma, coagulation biomarkers and their potential prognostic value.

Materials and methods section is descriptive enough as it refers and analyze the study's population, treatment protocol and follow-up procedures, data collection and statistical analyses implemented during the present study.

Results are, to my opinion, quite interesting and well depicted in relative tables and figures.

In discussion section the authors attempt to summarize and critical assess their main findings and to correlate them with relative literature data. They also highlight the main limitations of their study, a fact that I believe that further enhance the scientific value of this paper.

Conclusion section is well written. Despite that, the authors could aadd some sentences suggesting some specific targets for future studies.

Author Response

Dear Reviewer,

Thank you for your thoughtful and constructive feedback on our manuscript. We are grateful for your recognition of the strengths of our study, particularly the relevance of our findings and the clarity of our presentation.

In response to your suggestion regarding the inclusion of specific directions for future research, we have revised the conclusion section to address this point. We have expanded the discussion on the mechanistic relationship between coagulation pathways and glioblastoma progression, highlighting the importance of further research to elucidate how hypercoagulability influences tumor behavior. Additionally, we have incorporated a suggestion to investigate the impact of anticoagulant therapies on glioblastoma treatment outcomes, as this could offer new therapeutic insights. Finally, we emphasize the necessity of larger, prospective studies to validate our findings and assess the clinical applicability of the biomarkers identified in our study.

We believe that these additions improve the manuscript by providing a clearer path forward for future investigations. Once again, we greatly appreciate your insightful comments, which have helped refine and strengthen the manuscript.

Thank you for your time and valuable input.

Best regards.

Reviewer 2 Report

Comments and Suggestions for Authors

In this study, the relationship between coagulation parameters (such as D-dimer...) and survival rate of glioblastoma patients after radiotherapy was investigated. It is an interesting study, but it needs correction:

  1. In this study, test results are reported as lower or higher than normal, while higher values ​​indicate greater damage. Therefore, comparison of results may not be accurate enough. It would have been better if accurate results were reported as numbers and then grouped and compared.
  2. Regarding the effect of fibrinogen and platelet count: It would have been better if these tests had also been performed and compared.
  3. Figure 3 should be corrected, it is a repetition curve
  4. Table 1 should be completed, it cannot be seen completely

Author Response

Dear Reviewer,

We would like to thank the reviewer for their valuable comments and constructive feedback. Regarding the classification of coagulation biomarkers, we would like to clarify that, rather than categorizing the biomarkers as simply "higher" or "lower," we determined the optimal cut-off values using ROC curve analysis. This is a widely accepted approach in prognostic research, as it allows for the identification of the most informative thresholds. Based on these cut-off values, we then stratified the biomarkers into two groups and assessed their impact on progression-free survival (PFS) and overall survival (OS) using univariate Cox regression analysis.

In response to the reviewer’s comment, we have also incorporated pre-radiotherapy platelet count data into our analysis. We used ROC analysis to establish the optimal cut-off point for platelet count and evaluated its association with PFS and OS using univariate Cox regression. The updated results are included in the revised tables.

We would like to acknowledge the limitation regarding the absence of fibrinogen data in our study. Unfortunately, this important coagulation marker was not available for analysis. We have now explicitly mentioned this limitation in the revised manuscript and suggested that future studies should include fibrinogen measurements to better understand its potential prognostic role in glioblastoma.

Furthermore, we have addressed the reviewer’s comment regarding Figure 3 by removing the redundant figure, and we have completed Table 1 to provide a more comprehensive and clearer presentation of the data.

We hope that these revisions adequately address the reviewer’s concerns and enhance the clarity of our manuscript. We greatly appreciate the opportunity to improve our work and believe that these changes strengthen the study’s overall contribution.

Thank you again for your thoughtful feedback.

Reviewer 3 Report

Comments and Suggestions for Authors

In this study, the authors attempt to investigate the prognostic significance of blood coagulation markers, particularly prothrombin, thrombin, D-dimer, and albumin in glioblastoma diagnosis. The study is interesting and well-designed. However, there are major discrepancies that need to be addressed before the study can be published. The following are my concerns

  • The study highlighted the correlation of progression-free survival and D-dimer and prothrombin levels, in which, the lower the D-dimer level/ the higher the prothrombin level, the greater the survival. Rather than the glioblastoma prognostic, these patterns are more resemble the internal bleeding within the brain, either due to new trauma or previous surgery.
    1. For your information, during the acute blood coagulation, prothrombin convert into thrombin to stop the bleeding. The higher the prothrombin level, the lower the sign of internal bleeding in the brain. Similarly, D-Dimer is used to evaluate the presence of thrombosis / blood clots in blood vessels. The lower the D-Dimer, the lower the likelihood of incomplete blood clot. In summary, both prothrombin and D-dimer are the great prognostic markers in determining the surgery outcome, but there are not specific to glioblastoma.
  • Importantly, the study does not show data regarding glioblastoma tumors, either the CT-scan or MRI imaging data of tumor size within the brain or tumor proliferation markers (Ki67, PCNA etc…) from glioblastoma biopsy. These data are critical in correlating the tumor prognostic. With current data, it is difficult to correlate the blood coagulation markers with glioblastoma prognostic.

In light of these, I would like to recommend a revision for this manuscript, and I look forward to the revised manuscript.

Author Response

Dear Reviewer,

Thank you for your valuable comments and constructive feedback. We have made several revisions based on your suggestions and hope that the changes adequately address your concerns.

Tumor Volume and Correlations with Coagulation Biomarkers: We have now included data regarding tumor volume, determined optimal cut-off points for tumor volume using ROC analysis, and examined its effects on progression-free survival (PFS) and overall survival (OS) through Cox regression analysis. Additionally, we demonstrated the correlations between tumor volume and coagulation biomarkers using Spearman correlation analysis. The results from these analyses have been added to the relevant sections of the Results and Discussion, where we further elaborated on their implications.

Ki-67 Index: We also included the Ki-67 index data and performed separate Cox regression analyses for PFS and OS based on Ki-67. The results are now presented in the appropriate tables.

Prothrombotic Properties of Glioblastoma: The first paragraph of the Discussion section has been updated with references to two recent studies published in the last two years. These studies highlight the prothrombotic properties of glioblastoma and their relationship to prognosis. They provide further insight into the role of coagulation pathways in glioblastoma prognosis.

Bleeding Considerations: We recognize the importance of considering internal bleeding as a potential factor when interpreting coagulation biomarkers. In the Limitations section, we have added the following sentence: "Furthermore, although these coagulation biomarkers are significant in glioblastoma prognosis, it is important to consider that their patterns may indicate internal brain bleeding, possibly reflecting recent trauma or prior surgery."

We believe these revisions address the concerns raised in your review. We have carefully considered the importance of coagulation biomarkers as potential prognostic factors while also acknowledging the possibility of their association with brain hemorrhages. We hope these additions and clarifications strengthen the manuscript.

Thank you again for your thoughtful review, and we look forward to your further feedback.

Round 2

Reviewer 3 Report

Comments and Suggestions for Authors

The authors have address all the comments and make corrections and improvements in this manuscript. The manuscript is now ready for the publication.